# Fault Identification and Localization of a Time−Frequency Domain Joint Impedance Spectrum of Cables Based on Deep Belief Networks

**DOI:** 10.3390/s23020684

**Published:** 2023-01-06

**Authors:** Qingzhu Wan, Yimeng Li, Runjiao Yuan, Qinghai Meng, Xiaoxue Li

**Affiliations:** 1School of Electric and Control Engineering, North China University of Technology, Beijing100144, China; 2Key Account Division, Beijing Aerospace Data Stock Company National Big-Data Application Technology, Beijing100044, China

**Keywords:** deep belief network, cable fault location, fault type identification, headend input impedance spectrum, time–frequency domain

## Abstract

To improve the accuracy of shallow neural networks in processing complex signals and cable fault diagnosis, and to overcome the shortage of manual dependency and cable fault feature extraction, a deep learning method is introduced, and a time−frequency domain joint impedance spectrum is proposed for cable fault identification and localization based on a deep belief network (DBN). Firstly, based on the distribution parameter model of power cables, we model and analyze the cables under normal operation and different fault types, and we obtain the headend input impedance spectrum and the headend input time−frequency domain impedance spectrum of cables under various operating conditions. The headend input impedance amplitude and phase of normal operation and different fault cables are extracted as the original input samples of the cable fault type identification model; the real part of the headend input time–frequency domain impedance of the fault cables is extracted as the original input samples of the cable fault location model. Then, the unsupervised pre−training and supervised inverse fine−tuning methods are used for automatically learning, training, and extracting the cable fault state features from the original input samples, and the DBN−based cable fault type recognition model and location model are constructed and used to realize the type recognition and location of cable faults. Finally, the proposed method is validated by simulation, and the results show that the method has good fault feature extraction capability and high fault type recognition and localization accuracy.

## 1. Introduction

With the development of urbanization and the rapid growth of electricity demand, cross−linked polyethylene (XLPE) power cables have been widely used in power grids because of their good electrical and mechanical properties, visual aesthetics, and substantial savings in land resources [1,2]. The power cables in urban power supply systems are laid in cable trenches or buried directly underground, and due to external damage, poor construction, corrosion of the protective layer, etc., the local position of the cable is easily affected and the electrical parameters at this position change relative to the normal value, forming local defects. Since power cables are mostly laid underground, it is difficult to directly detect defective points, and these defects will gradually develop under the action of electric fields, leading to cable faults such as open circuits, short circuits, high resistance, low resistance, etc. [3,4]. However, common diagnostic methods have disadvantages such as the inability to determine the type of fault and low localization accuracy, and they cannot manually identify a large amount of cable fault information in a short period of time. Research on cable fault identification and localization can avoid causing significant economic losses and diagnose power systems in time before permanent faults occur, in order to improve their operational reliability.

Currently, common cable fault detection methods include time domain reflectometry (TDR), frequency domain reflectometry (FDR), broadband impedance spectroscopy (BIS), etc. The TDR method determines the location of cable faults by transmitting a pulse signal at the beginning or end of the faulty cable and calculating the time difference between the incident and reflected signals [5]. The authors of [6,7], based on the TDR method, emitted compressed pulses to detect the status of cable joints, but the waveforms obtained could only determine open and short circuit faults, without high resistance and low resistance faults. Additionally, the accuracy of the cable defect location was low, due to the low content of high−frequency components of the pulse signal. The FDR method uses a swept sinusoidal signal as the incident signal of the cable and determines whether a local defect occurs in the cable based on the distortion characteristics of the reflected signal spectrum of the cable. In [8,9,10], based on the FDR method, swept−frequency signals were used to diagnose cable defects, and the method was more accurate in localization because the signal contained more high−frequency energy, but it failed to identify different types of cable faults. Meanwhile, single domain analysis faces difficulties in diagnosing cable faults and defects, due to the resolution limitation and weak anti−interference capability of reflected signals. The BIS method uses transmission line theory to determine cable defects by comparing the differences in the broadband impedance spectrum of cables before and after aging [11]. The authors of [12,13,14] used the cable broadband impedance spectrum processed by the BIS method to identify cables’ thermal aging defects, but the BIS method was only applied to cable defect diagnosis and did not identify cable faults such as open circuits and short circuits. The authors of [15,16,17] extracted fault features such as power waveforms, zero−sequence currents, and cable sheath currents’ abrupt change information, respectively, and used deep learning algorithms to construct classification models to identify fault types; the fault features extracted by the above methods required further calculations, and the fault diagnosis efficiency was low, the feature extraction process was more complicated, and no research was carried out on fault localization.

To identify large amounts of cable fault information more efficiently and accurately, this paper proposes a fault identification and localization method using a time–frequency domain joint impedance spectrum of cables based on deep belief networks. Using the strong generalization ability and autonomous feature extraction capability of DBNs, this method obtains the headend input impedance spectrum of the cable by building cable models under normal operation and fault conditions, and then it uses the headend input impedance amplitude and phase of normal operation and fault cables as the original samples for training and testing the cable fault type identification model, obtains the headend input time–frequency domain impedance spectrum of the cables based on inverse fast Fourier transform (IFFT), and uses the real part of the headend input time–frequency domain impedance of different faults as the original samples for training and testing the cable fault type identification model. The real part of the frequency domain impedance at the headend input of the fault is used as the original sample for training and testing the cable fault location model. The unsupervised pre−training and supervised inverse fine−tuning methods are used to automatically learn, train, and extract the cable fault state features from the original samples and construct the DBN−based cable fault type identification model and localization model, respectively, so as to achieve the cable fault type identification and localization. Finally, the effectiveness of the proposed method is verified by simulation. The simulation results show that the method has good fault feature extraction capability, high fault type identification, and high localization accuracy, which can be extended to the practical applications of smart grids combined with the situation.

## 2. Fault Cable Impedance Spectrum Identification and Location Principle

### 2.1. Fault Cable Distribution Parameter Equivalent Circuit

According to transmission line theory, under a high−frequency power supply, power cables can be equivalent to a distributed parameter model for describing the transmission characteristics of signals in power cables, which are characterized by the input impedance. The cable containing the fault point C is equated to the following model, as shown in Figure 1b, where *R*, *L*, *C*, and *G* are the unit resistance (Ω/m), unit inductance (H/m), unit capacitance (F/m), and unit conductance (S/m) of the cable, respectively.

From reference [18], it is known that under high frequency, with a cable of length *l*, at the transmission line, the voltage of position C ${\dot{V}}_{CB}(l_{f})$ and its current ${\dot{I}}_{CB}(l_{f})$ are as follows:(1)$$\left\{ \begin{array}{l} {\dot{V}}_{CB}(l_{f})={\dot{V}}^{+}e^{\gamma (l-l_{f})}+{\dot{V}}^{-}e^{-\gamma (l-l_{f})}\\ {\dot{I}}_{CB}(l_{f})=\frac{1}{Z_{0}}({\dot{V}}^{+}e^{\gamma (l-l_{f})}-{\dot{V}}^{-}e^{-\gamma (l-l_{f})}) \end{array}\right.$$
where ${\dot{V}}^{+}$ is the incident voltage wave propagating in the direction of the load, ${\dot{V}}^{-}$ is the reflected voltage wave propagating in the direction of the power supply, and *l_f_* is the length of point C from the headend of the cable.

The voltage of position B ${\dot{V}}_{AB}(l)$ and its current ${\dot{I}}_{AB}(l)$ are as follows:(2)$$\left\{ \begin{array}{l} {\dot{V}}_{AB}(l)={\dot{V}}^{+}e^{\gamma l}+{\dot{V}}^{-}e^{-\gamma l}\\ {\dot{I}}_{AB}(l)=\frac{1}{Z_{0}}({\dot{V}}^{+}e^{\gamma l}-{\dot{V}}^{-}e^{-\gamma l}) \end{array}\right.$$

The characteristic impedance *Z*_0_ of the cable is
(3)$$Z_{0}=\sqrt{\frac{R+j\omega L}{G+j\omega C}}$$

At high frequency, *ωL* >> *R*, *ωC* >> *G*; the above equation can be approximated as follows:(4)$$Z_{0}=\sqrt{\frac{L}{C}}$$

The propagation coefficient *γ* is
(5)$$\gamma =\sqrt{\left(R+j\omega L)(G+j\omega C\right)}=\alpha +j\beta$$
where *α* is the decay constant, characterizing the amplitude decay characteristics of the voltage and current waves.
(6)$$\alpha =\frac{R}{2}\sqrt{\frac{L}{C}}$$*β* is the phase constant, characterizing the phase−change characteristics of the voltage and current waves.
(7)$$\beta =\frac{\omega }{v}=\frac{2\pi f}{v}=\frac{2\pi }{\lambda }=\omega \sqrt{LC}$$
where *v* is the propagation speed of the signal wave in the cable, *v* is a constant at high frequency, *f* is the frequency, and *λ* is the wavelength of the transmitted signal.

The reflection coefficient *Г*_L_ of the cable terminal load *Z*_L_ is
(8)$$\Gamma_{L}=\frac{{\dot{V}}^{-}}{{\dot{V}}^{+}}=\frac{Z_{L}-Z_{0}}{Z_{L}+Z_{0}}$$

If the cable load is open−circuit (*Z*_L_ = ∞), then *Г*_L_ = 1; if the cable load is short−circuit (*Z*_L_ = 0), then *Г*_L_ = −1; when there is no fault (*Z*_L_ = *Z*_0_), then *Г*_L_ = 0. In this paper, we set the cable’s terminal load as open−circuit.

### 2.2. Cable Impedance Spectrum Fault Identification Principle

As shown in Figure 1a, in normal operation, the transmission signal is reflected at the terminal point B. The expression of the cable headend input impedance is as follows, which can be obtained by dividing ${\dot{V}}_{AB}(l)$ by ${\dot{I}}_{AB}(l)$:(9)$$Z_{AB}(l)=\frac{{\dot{V}}_{AB}(l)}{{\dot{I}}_{AB}(l)}=Z_{0}\frac{1+\Gamma_{{}_{L}}e^{-2\gamma l}}{1-\Gamma_{{}_{L}}e^{-2\gamma l}}$$

Substituting the propagation coefficient *γ* = α + *jβ* into the headend input impedance yields
(10)$$Z_{AB}(l)=Z_{0}\frac{1+\Gamma_{L}e^{-2\alpha l}e^{-2j\beta l}}{1-\Gamma_{L}e^{-2\alpha l}e^{-2j\beta l}}$$

If the cable fails at C, the transmitted signal is folded back at point C. The input impedance *Z*_CB_(*l_f_*) at this point is as follows, which can be obtained by dividing ${\dot{V}}_{CB}(l_{f})$ by ${\dot{I}}_{CB}(l_{f})$:(11)$$Z_{CB}(l_{f})=\frac{{\dot{V}}_{CB}(l_{f})}{{\dot{I}}_{CB}(l_{f})}=Z_{0}\frac{1+\Gamma_{L}e^{-2\gamma (l-l_{f})}}{1-\Gamma_{L}e^{-2\gamma (l-l_{f})}}$$

The expression of the cable input impedance at the headend is
(12)$$Z_{AC}(l_{f})=Z_{0}\frac{1+\Gamma_{L1}e^{-2\gamma l_{f}}}{1-\Gamma_{L1}e^{-2\gamma l_{f}}}$$

The reflection coefficient of the fault point C is
(13)$$\Gamma_{L1}=\frac{(Z_{CB}//R_{f})-Z_{0}}{(Z_{CB}//R_{f})+Z_{0}}$$
where *R_f_* is the equivalent resistance of the fault point C. The impedance of the CB section can be used as the load impedance of the AC section by connecting it in parallel with *R_f_* [18].

The frequency change curve of the headend impedance is called the cable headend input impedance spectrum, including the impedance amplitude spectrum and the phase spectrum, which is a function of the reflection point position, the propagation coefficient, and its characteristic impedance. For the cable, due to the frequency dependence of its distribution parameters, the cable transmission characteristics change when a fault occurs in the cable, and the cable headend input impedance spectrum changes with the change in the power supply frequency. Therefore, the input impedance spectrum can characterize the state information at different cable locations, and its frequency spectrum can reflect the transmission characteristics of the cable. Compared a faulty cable input impedance spectrum and a normal cable, the characteristic differences in the spectrum can be used to identify the type of cable fault.

### 2.3. Cable Time–Frequency Domain Impedance Spectrum Fault Location Principle

The transmission state of the signal in the cable is considered to be a traveling wave transmission. When a fault occurs in the cable, the transmitted signal is folded and reflected at the fault point. Using this transmission characteristic, we introduced the time–frequency domain reflection method to locate cable faults, as shown in Figure 1a.

From the cable input impedance spectrum Equations (9) and (12), it can be seen that the input impedance spectrum can characterize the cable fault type, but it cannot directly characterize the reflection point location, so it needs to be transformed to the time domain to obtain the time–frequency domain impedance spectrum, and the headend input time–frequency domain impedance spectrum is obtained by the signal time–frequency domain analysis. The real part of the time–frequency domain impedance spectrum is extracted for localization, which characterizes the fault location, and the method has a high accuracy of localization.

The IFFT of the headend input impedance of the cables under normal operation and fault conditions can be obtained using Equations (14) and (15), respectively:(14)$$Z_{AB}(t)=\frac{1}{2\pi }\int_{-\infty }^{+\infty }Z_{AB}(l)e^{j\omega t}d\omega$$
(15)$$Z_{AC}(t)=\frac{1}{2\pi }\int_{-\infty }^{+\infty }Z_{AC}(l_{f})e^{j\omega t}d\omega$$

By finding the peak points in the headend input time–frequency domain impedance spectrum, the location of the cable fault point and the terminal of the cable can be determined, which can determine the cable fault location.

## 3. Establish Sample Database

### 3.1. Collection of Sample Data

For the common types of cable faults, cable models under normal operation and open circuit, short circuit, high resistance, and low resistance faults under five operating conditions were established, respectively. By randomly combining different fault locations and ground transition resistances, we obtained different operating conditions for the cable headend input impedance spectrum and their time–frequency domain impedance spectrum, with a sampling frequency of 0–50 MHz, each fault sampling 1001 points, and the cable terminal set to an open circuit state. In this study, the cable headend input impedance amplitude and phase for normal operation and different faults were extracted as the original samples for the cable fault type identification model; the real part of the cable headend input time–frequency domain impedance for different faults was extracted as the original samples for the cable fault location model.

### 3.2. Sample Database Generation for Fault Type Identification and Localization

Based on Equation (9), the cable headend input impedance amplitude and phase under normal operation are
(16)$$\left\{ \begin{array}{l} {\left|Z_{AB}(l)\right|}_{i}=\left|Z_{0}\right|\sqrt{\frac{1+r^{2}+2r\cos\frac{4\pi f}{v}l}{1+r^{2}-2r\cos\frac{4\pi f}{v}l}}\\ \varphi ({Z_{AB}}_{)i}=\varphi (Z_{0})-arctan\frac{2rsin\frac{4\pi f}{v}l}{1-r^{2}} \end{array}\right.$$
where r = *Г*_L_e^−2α^*^l^*, *i* = 1, 2,…, *m*. |*Z*_AB_(*l*)|*_i_* and φ(*Z*_AB_)*_i_* are the *i*th cable headend input amplitude and phase under normal conditions, respectively.

Based on Equation (12), the cable headend input impedance amplitude and phase under different faults are
(17)$$\left\{ \begin{array}{l} {\left|Z_{AC}(l_{f})\right|}_{j}=\left|Z_{0}\right|\sqrt{\frac{1+r_{1}{}^{2}+2r_{1}cos\frac{4\pi f}{v}l_{f}}{1+r_{1}{}^{2}-2r_{1}cos\frac{4\pi f}{v}l_{f}}}\\ \varphi ({Z_{AC}}_{)j}=\varphi (Z_{0})-arctan\frac{2r_{1}sin\frac{4\pi f}{v}l_{f}}{1-r_{1}{}^{2}} \end{array}\right.$$
where $r_{1}=\Gamma_{L1}e^{-2\alpha l_{f}}$, *j* = 1, 2, …, *n*. |*Z*_AC_(*l_f_*)|*_j_* and φ(*Z*_AC_)*_j_* are the *j*th cable headend input amplitude and phase under different faults, respectively.

The headend input impedance amplitude and phase under normal operation and different faults can be taken to generate the sample vector as follows:(18)$$Z_{AB}{}_{i}=[{\left|Z_{AB}(l)\right|}_{i},\varphi {\left(Z_{AB}\right)}_{i}]$$
(19)$$Z_{AC}{}_{j}=[{\left|Z_{AC}(l_{f})\right|}_{j},\varphi {\left(Z_{AC}\right)}_{j}]$$
where ***Z***_AB*i*_ and ***Z***_AC*j*_ are the sample vectors of the *i*th normal operating cable and the *j*th faulty cable, respectively.

The input sample database of the cable fault type identification model is
(20)$$Z={\left[Z_{AB}{}_{1},Z_{AB}{}_{2},\ldots ,Z_{AB}{}_{i},\ldots ,Z_{AB}{}_{m},Z_{AC}{}_{1},Z_{AC}{}_{2},\ldots ,Z_{ACj},\ldots ,Z_{AC}{}_{n}\right]}^{T}$$

The real part of the faulty cable’s headend input time–frequency domain impedance is
(21)$$real{\left(Z_{AC}(t,f)\right)}_{j}=\left|Z_{AC}(l_{f})\right|\cos(2\pi tf)$$
where *j* = 1, 2,…, *n*, and *real*(*Z*_AC_(*t*,*f*))*_j_* is the *j*th cable’s real part of the different faulty cables’ headend input time–frequency domain impedance.

The real part of the different faulty cables’ headend input time–frequency domain impedance can be taken to generate the sample vector as follows:(22)$$R={\left[real{\left(Z_{AC}(t,f)\right)}_{1},real{\left(Z_{AC}(t,f)\right)}_{2},\ldots ,real{\left(Z_{AC}(t,f)\right)}_{j},\ldots ,real{\left(Z_{AC}(t,f)\right)}_{n}\right]}^{T}$$

The sample database for fault type identification and localization is generated using the matrices ***Z*** and ***R***, respectively, and the sample database is randomly divided into a training sample set and a test sample set at a 4:1 ratio.

## 4. A Deep Belief Network−Based Model for Cable Fault Type Identification and Location

### 4.1. Principle of the Deep Belief Network

DBNs can effectively learn the internal features of cable fault type identification and localization samples with strong generalization ability and autonomous feature extraction [19], and this paper proposes a DBN−based cable fault type identification and localization method to automatically learn and extract fault state information from the original samples to achieve cable fault type identification and localization [20,21,22]. According to the different functions, the DBN−based cable fault classification model and its localization model were established. Among them, the classification model is for discrete data, which are used for fault type identification of open circuit, short circuit, high resistance, and low resistance faults, while the localization model is for continuous data, which are used for the location of cable fault points.

### 4.2. Structure and Training Process of the Deep Belief Network

The structure of the DBN−based cable fault type identification and localization model consists of multiple stacked restricted Boltzmann machines (RBMs) in the bottom layer and an error backpropagation (BP) network in the top layer. The training process is divided into layer−by−layer pre−training with unsupervised learning and backward fine−tuning with supervised learning, as shown in Figure 2 [23].

The unsupervised layer−by−layer pre−training of the DBN can be considered as the training of multiple independent RBMs. Through the layer−by−layer stacking of RBMs, the entire network can effectively extract features from the headend input impedance amplitude and phase sample database ***Z*** and the headend input time–frequency domain impedance real part sample database ***R*** layer by layer, greatly decreasing the learning difficulty of the fault diagnosis model and accurately restoring the cable fault type and its location characteristics [24].

The pre−training results tend to converge to the local optimal solution and need to be optimized in reverse to achieve convergence to the global optimal solution. The top layer of the DBN is added with the classification labels of the cable operating conditions and the fault location labels, and the weights and biases are fine−tuned for each layer of the fault type identification and localization network by supervised backward fine−tuning to improve the feature extraction capability of the model for cable faults and reduce the training error of the fault type identification and localization model [23].

### 4.3. Process of DBN−Based Cable Fault Diagnosis

The DBN−based cable fault diagnosis method has three stages, including data pre−processing, model training, and identifying faults. For the problem of this paper, we established two models for fault identification and fault location, respectively, as shown in Figure 3. The specific steps are as follows:

Step 1The headend input impedance amplitude and phase of the normal operation and faulty cables are extracted as the original sample database ***Z*** of the fault identification model, and the real part of the faulty cable’s headend input impedance time–frequency domain is extracted as the original sample database ***R*** of the fault location model.Step 2Data pre−processing of fault samples, including normalizing the samples and dividing them into a training set and test set according at a 4:1 ratio.Step 3Building a DBN−based cable fault type identification and localization model, the unlabeled training sample set is input to the RBM containing three hidden layers for unsupervised learning.Step 4Add softmax classifier, encode the fault type and fault location, and set the number of nodes in the output layer to 10.Step 5Inverse fine−tuning of the model; using the labeled samples, the pre−trained cable fault diagnosis feature parameters are subjected to supervised inverse fine−tuning using a BP neural network, which makes the fault diagnosis network’s performance converge to the global optimum, and the root−mean−square error (*RMSE*) is used as the training error to evaluate the network performance, the expression of which is as follows:
(23)$$RMSE=\sqrt{\frac{\sum_{i=1}^{N}{\left(Z_{pre}^{i}-Z_{true}^{i}\right)}^{2}}{N}}$$
where *Z^i^_pre_* and *Z^i^_true_* are the predicted and actual faults under the *i*th fault condition in the BP neural network, respectively, and N denotes the total number of samples. The end conditions of the inverse fine−tuning are set as follows: the *RMSE* is less than the preset value of 0.01, and the number of iterations reaches the set 1000. The smaller the training error, the better the fit of the DBN model to the training sample set.Step 6Test model—the test sample set is input to the DBN model for fault prediction, and the prediction results are compared with the actual fault conditions to evaluate the performance of the two DBN models trained.

## 5. Simulation of DBN−Based Cable Fault Identification and Location

### 5.1. Construction of Cable Fault Model

According to the common cable faults, divided into open circuit, short circuit, high resistance, and low resistance faults, the equivalent resistance *R_f_* at the cable fault points corresponding to different fault types is shown in Table 1.

We built a 10 kV single−core XLPE cable fault simulation model based on PSCAD simulation to obtain fault sample data. The simulation cable model was set to 100 m length, the sampling frequency was 0–50 MHz, each fault sampled 1001 points, and the end of the cable was set to an open circuit state.

### 5.2. Comparison of the Headend Input Impedance Spectrum of Normal and Faulty Cables

Figure 4 shows the simulated headend input impedance amplitude spectrum for a 100 m cable with open and short circuit faults at 50 m. From the simulation results, it can be seen that both open circuit faults and short circuit faults lead to a decrease in amplitude and a larger resonant period compared to the normal operation cable.

Figure 5 shows the simulated amplitude spectrum of the headend input impedance for a 100 m cable with high and low resistance faults at 50 m. From the simulation results, it can be seen that the high and low resistance faults lead to amplitude attenuation. The low resistance fault also leads to an increase in the resonant period compared to the normal operation cable.

Figure 6 shows the simulated phase spectrum of the headend input impedance for a 100 m cable with open circuit and short circuit faults at 50 m. From the simulation results, it can be seen that an open circuit fault leads to a larger resonant period and a larger initial phase compared to the normal operation cable, while a short circuit fault leads to a larger resonant period and a positive initial phase.

Figure 7 shows the simulated phase spectrum of the headend input impedance for a 100 m cable with high and low resistance faults at 50 m. From the simulation results, it can be seen that compared with the normal operation cable, the high resistance fault leads to a smaller initial phase and phase decay, while the low resistance fault leads to an increased resonant period, positive initial phase, phase decay, and resonance point shift.

From Figure 4 to Figure 7, it can be seen that both the headend input impedance amplitude spectra and phase spectra of the cable decrease periodically with the increasing frequency, and the maximum value points of the impedance amplitude spectra appear periodically, while the zero−points of the phase spectra appear periodically. The lines near the maximum value points of the amplitude spectra and the zero−points of the phase spectra are very steep, which can reflect the transmission characteristics of the cables sensitively.

From the cable in the high−frequency equivalent circuit of the distribution parameters shown in Figure 1b, it can be seen that with the increase in the frequency, the cable will display series resonance periodically, and the resonant frequency points appear near the maximum value points of the impedance amplitude spectra and the zero−points of the phase spectra.

### 5.3. Location of Cable Faults

After the IFFT process—the real part of the normal operation and faulty cables’ input impedance simulated in Section 5.2—we obtained the location spectra of the open circuit, short circuit, high resistance, and low resistance faults in the 100 m cable at 50 m, as shown in Figure 8. According to the processing results, it can be seen that the headend input time–frequency domain impedance of the real part of the spectrum has two peak points at 1650 ns and 3300 ns, from which we can derive the corresponding cable lengths of 50 m and 100 m, respectively, by the distance formula *l* = *v·t*. The positioning accuracy is 100%, which is consistent with the fault location and the cable end set in the simulation, indicating that by transforming the real part of the headend input time–frequency domain impedance and finding the peak point in the time domain of the corresponding spectrum, the location of the cable fault point and the end of the cable can be determined, from which we can determine the cable fault’s location.

### 5.4. Simulation Analysis

In order to obtain enough training and test data, the fault data were collected by changing the fault location and ground transition resistance, and the fault conditions were set as follows: (a) fault location—a cable fault was set every 2 m between 4 m and 96 m; (b) ground transition resistance—5 Ω, 10 Ω, 20 Ω, 50 Ω, 100 Ω, 1000 Ω, 2000 Ω, 3000 Ω 4000 Ω, and 5000 Ω; a total number of 686 sets of samples were obtained, as shown in Table 2. For the fault classification model, the sample database ***Z*** consists of normal operation and fault headend input impedance amplitude and phase spectra, with 2002 points per sample, forming a total number of 686 2002−dimensional sample spaces; for the fault location model, the sample database ***R*** consists of the real part of the headend input time–frequency domain impedance at different fault points, with 1001 points per sample, forming a total number of 615 1001−dimensional sample spaces. In this study, we selected group *i* cable data in normal condition and group *j* data in fault condition, where *i* = 1, 2,…, 71 and *j* = 1, 2,…, 615. Then, 80% of the samples were selected as training data, and 20% of the samples were used as test data. The fault type and fault location were coded sequentially, as shown in Table 3 and Table 4, respectively.

### 5.5. DBN Model Fault Diagnosis Result Analysis

As can be seen from Figure 9, the error gradually decreased with the increase in the number of iterations, and the classification accuracy of the training set gradually increased. When the number of iterations reached about 300, the error was close to 0, and the classification accuracy of the training set reached 99.81%.

The accuracy of the DBN fault type recognition model was tested using the test dataset, and the results are shown in Figure 10. We achieved 99.27% prediction accuracy for the DBN test set, so the fault classification method proposed in this paper was able to accurately identify the fault type.

As can be seen from Figure 11, the error gradually decreased with the increase in the number of iterations, and the training set localization accuracy gradually increased. When the number of iterations reached about 250, the error was close to 0, and the training set localization accuracy reached 100%.

The accuracy of the DBN fault location model was tested using the test dataset, and the results are shown in Figure 12. The prediction accuracy of the DBN test set reached 100%, so the fault location method proposed in this paper was able to accurately locate the fault.

### 5.6. Comparative Analysis

In order to further analyze the effectiveness of the proposed method, the cable data were input into support−vector machine (SVM), artificial neural network (ANN), long short−term memory (LSTM), and convolutional neural network (CNN) models to learn and test the samples ***Z*** and ***R***. For SVM, the penalty factor was set to three, and the kernel function was chosen as a radial basis function (RBF); for the ANN, it adopted the same 1001−100−10 hidden layer structure as the DBN, and the training method used was a BP neural network; for LSTM, the number of hidden layer cells was set to 100, and the maximum number of cycles was set to 100; for the CNN, the number of convolutional layers was set to five, the number of pooling layers was set to three, and maximum pooling was adopted as the pooling method. All of the abovementioned methods divided the cable data into training and test sets at a 4:1 ratio, and the training end conditions used were as follows: the maximum number of iterations was 1000, and the root−mean−square function was used for the training error, which was less than 0.01. The fault identification and location results and the performance comparison of the five models are shown in Table 5.

Comparing the five models, it can be seen that the DBN model has the most practical applications. The LSTM, ANN, and SVM models produce significant misclassification of the sample data. Although the DBN model has a longer training time, its accuracy rate is significantly improved. The CNN model shows slightly better classification accuracy compared with the DBN model, but it needs a far longer training time. The DBN model can extract deeper hidden features from the original sample set and has better feature extraction capability. In this way, the effectiveness and practicality of the DBN−based cable fault type identification and localization method proposed in this paper is verified.

## 6. Conclusions

Aiming at the existing cable fault diagnosis methods with low accuracy and complex diagnosis methods, this paper applies deep learning to cable fault diagnosis and proposes a fault identification and localization method based on the time–frequency domain joint impedance spectrum of cables using deep belief networks. This method has a good fault feature extraction capability, as well as high fault type identification and localization accuracy.

(1)By modeling different types of cable operation with IFFT transformation, the headend input time–frequency domain impedance spectrum of the normal operation and different faulty cables were obtained as the original input samples of the DBN, and the performance of the model was analyzed by the fault type identification results of the DBN network and its localization results.(2)The simulation results showed that the DBN−based cable fault type identification and location method could maintain the original characteristics of the data in the process of data dimensionality reduction, and the fault identification and location results were unaffected by the fault location, transition ground resistance, and other factors. The fault identification and location accuracies reached 99.27% and 100%, respectively.(3)The method was able to effectively identify the type of cable fault and locate the fault points, which could be extended to practical applications for smart grids.

## Figures and Tables

**Figure 1 sensors-23-00684-f001:**
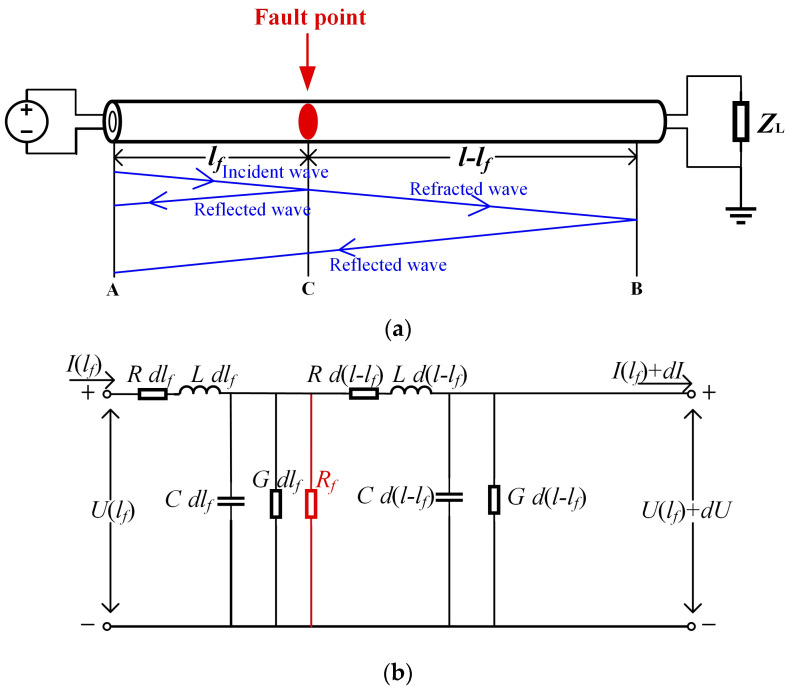
(**a**) High−frequency cable with fault point. (**b**) Fault cable in high−frequency equivalent circuit of distribution parameter.

**Figure 2 sensors-23-00684-f002:**
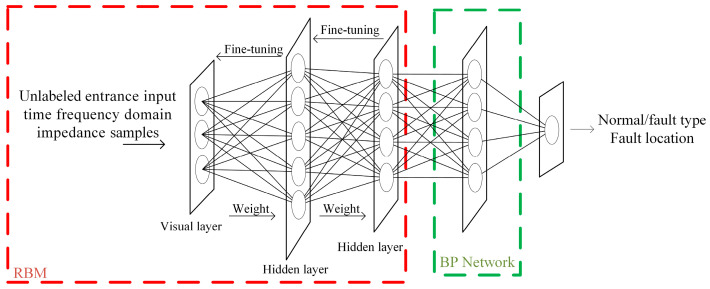
Basic structure of the DBN model.

**Figure 3 sensors-23-00684-f003:**
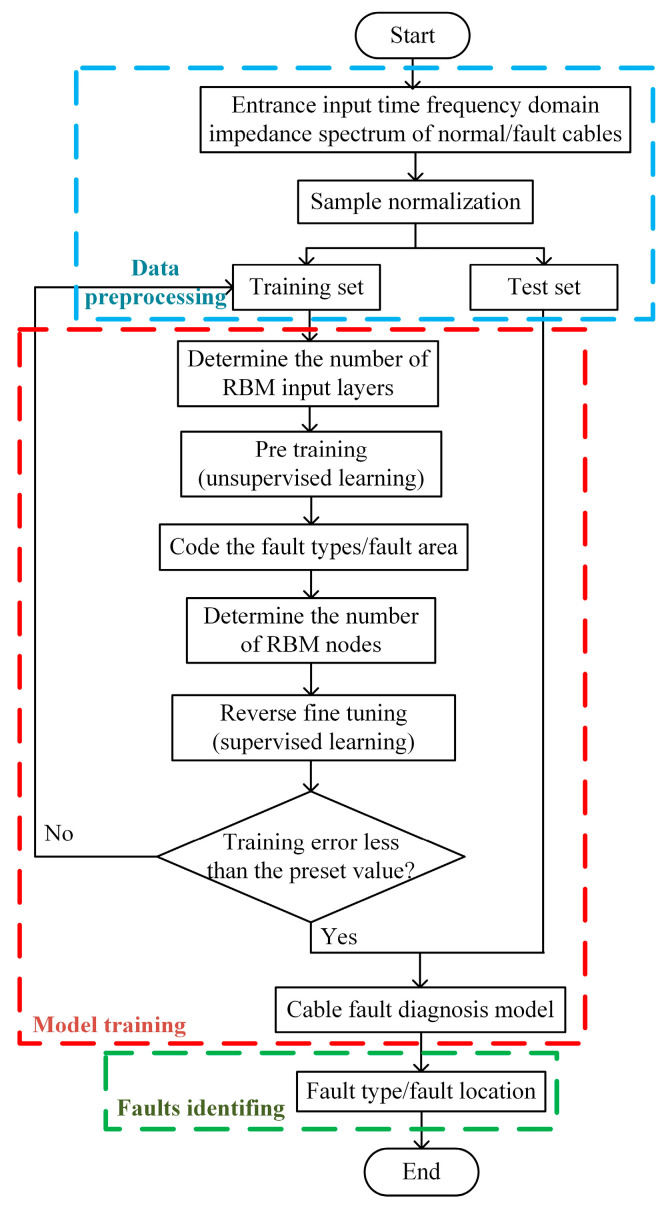
Cable fault diagnosis process based on DBN.

**Figure 4 sensors-23-00684-f004:**
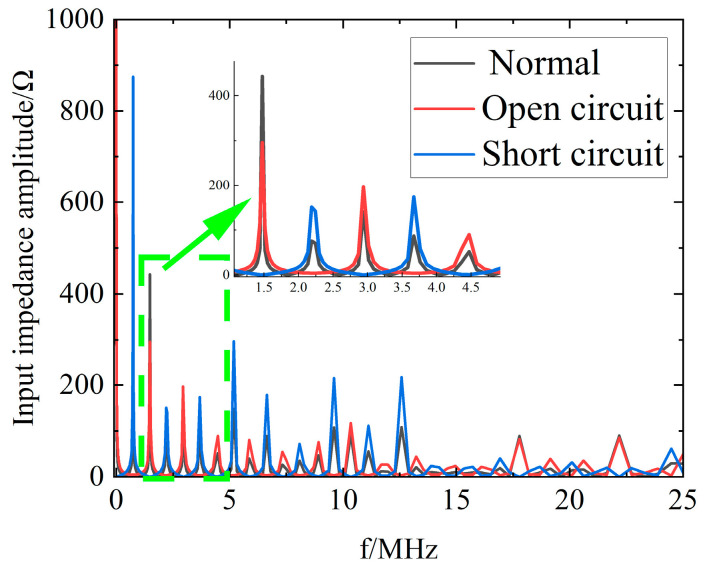
Headend input impedance amplitude spectrum of normal operation, open circuit, and short circuit cables.

**Figure 5 sensors-23-00684-f005:**
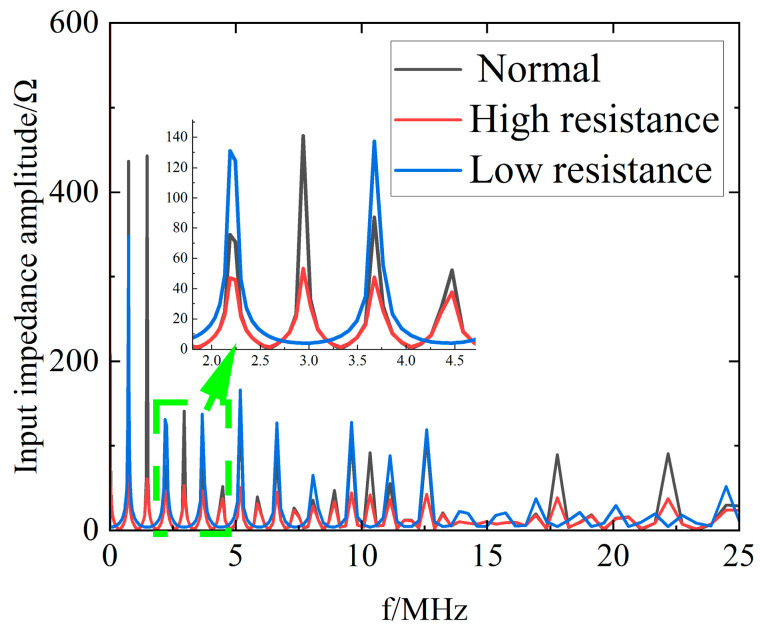
Headend input impedance amplitude spectrum of normal operation, high resistance, and low resistance cables.

**Figure 6 sensors-23-00684-f006:**
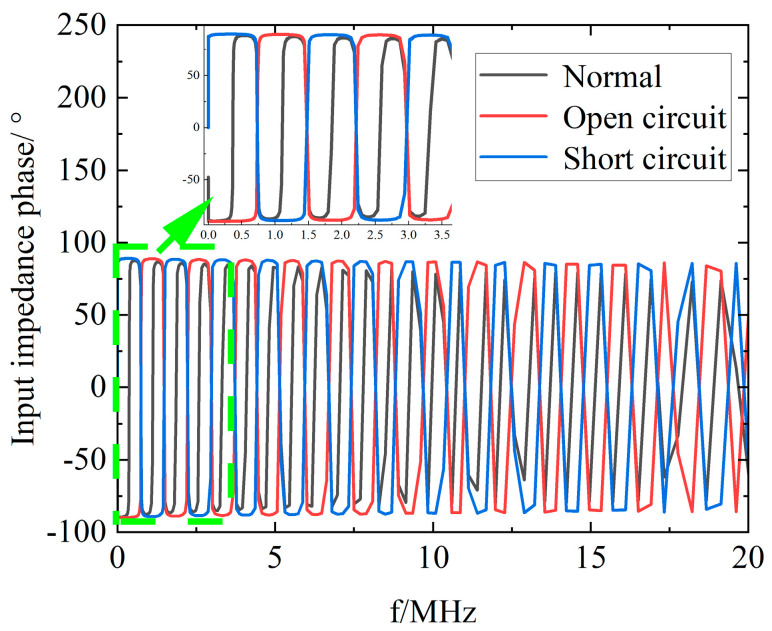
Headend input impedance phase spectrum of normal operation, open circuit, and short circuit cables.

**Figure 7 sensors-23-00684-f007:**
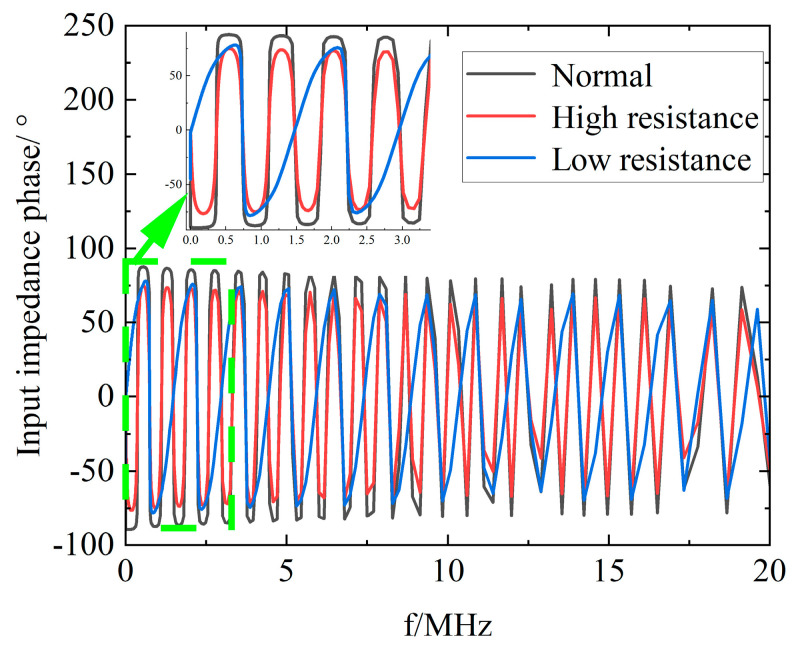
Headend input impedance phase spectrum of normal operation, high resistance, and low resistance cables.

**Figure 8 sensors-23-00684-f008:**
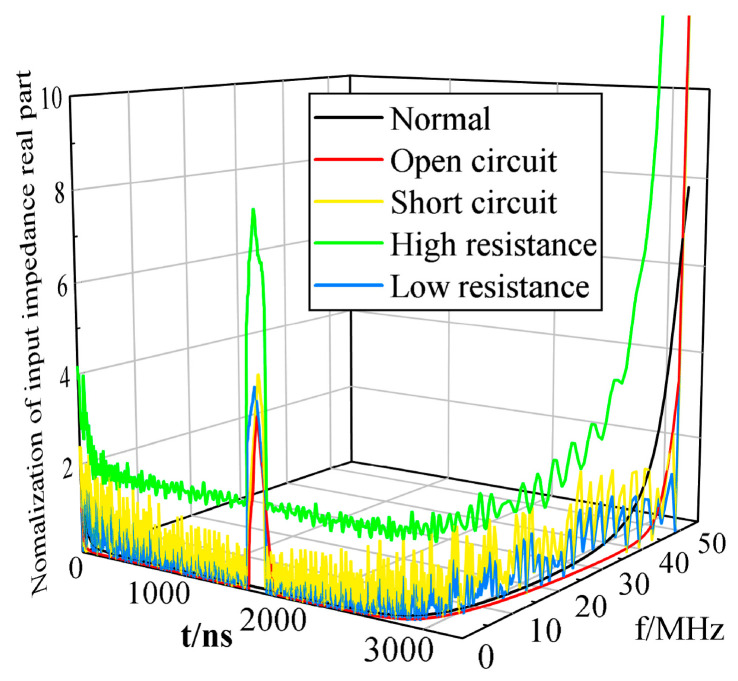
Headend input time–frequency domain impedance of the real part of the spectrum of normal operation and fault cables.

**Figure 9 sensors-23-00684-f009:**
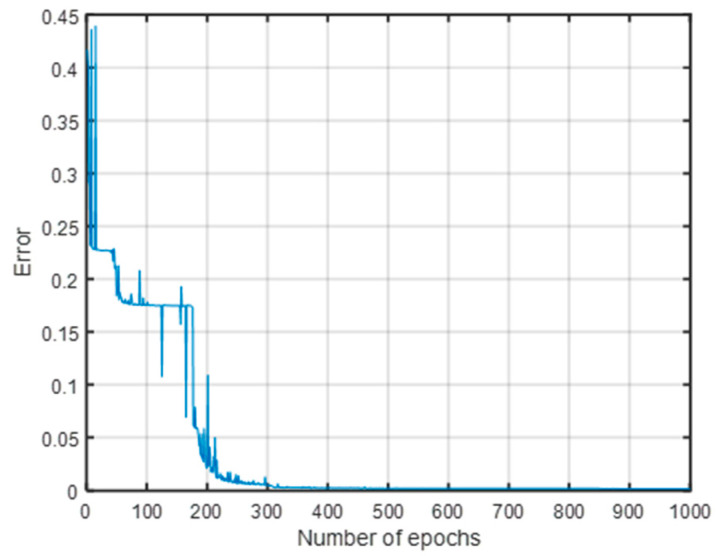
Error to iterations of the fault classification training set.

**Figure 10 sensors-23-00684-f010:**
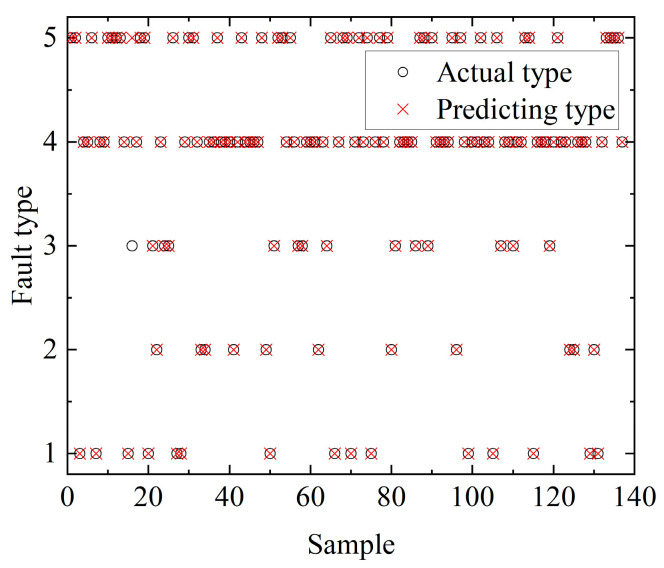
Test results of the DBN classification test set.

**Figure 11 sensors-23-00684-f011:**
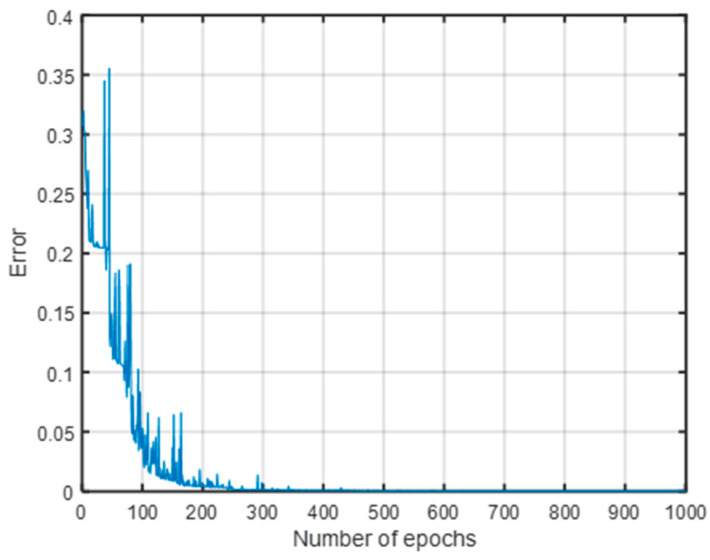
Error to iterations of the fault location training set.

**Figure 12 sensors-23-00684-f012:**
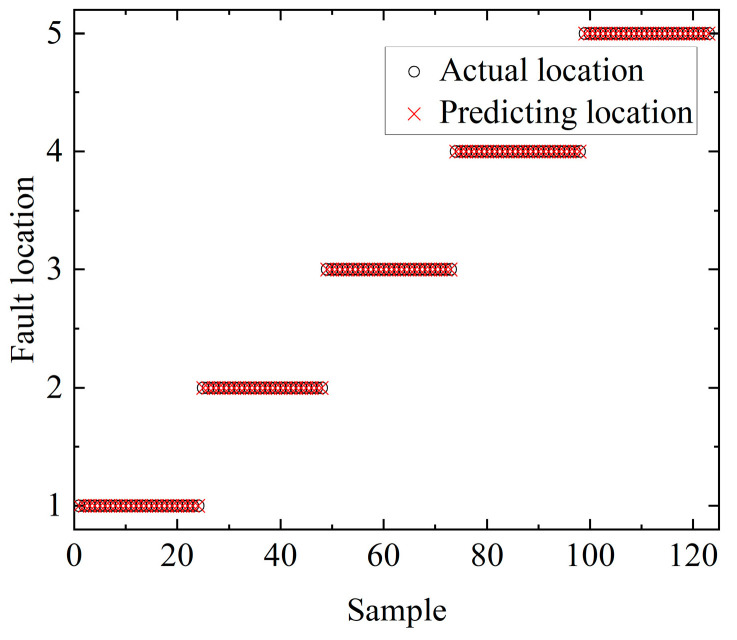
Test results of the DBN location test set.

**Table 1 sensors-23-00684-t001:** *R_f_* values of different types of cable faults.

Fault Type	Open Circuit	Short Circuit	High Resistance	Low Resistance
R*_f_*	∞	0	>10 *Z*_0_	<10 *Z*_0_

**Table 2 sensors-23-00684-t002:** Part of the sample database.

Operation Conditions	Sample Database	
	Fault Type Identification Sample Z	Fault Location Sample R
Normal	[1701547019,1147256261,…, 26.10192758,−47.60935,…, 53.32]	−
	[2041856423.2, 1992169260.7,…, 2.38349,−48.20147,…, 71.7711]	
	⁝	
	[1276160264, 926442784.9,…, 29.89945885,−51.12209,…, 49.30311]	
Fault	[134698812.7,34640942.7,…9.611459666, 1.307308543, −90,…, −82.61954]	[0.998035,0.64874,…, 3.11328,…, 3.527929, 46.09701]
	[0.202491386,114.0169151,…, 0.896704823,66.66257433,0.0058,…, −61.17]	[2.39336,1.58306,…, 4.41899,…, 0.12737,39.5892]
	[2871.114899,2665.930828,…, 18.24457638,1.53203625, −44.996,…, 33.94]	[4.16462,3.87678,…, 7.00472,…, 162.04205,314.36995]
	⁝	⁝
	[51.46603098,48.55095511,…, 22.82071183,30.27926315,−45,…, −49.02]	[0.90036,0.34598,…, 3.20213,…, 0.01832,44.97257]

**Table 3 sensors-23-00684-t003:** Sample volume and code of cable fault types.

Operation Conditions	Sample Data Size	Identification Tags
Normal	71	1
Open circuit fault	73	2
Short circuit fault	71	3
High resistance fault	235	4
Low resistance fault	235	5

**Table 4 sensors-23-00684-t004:** Sample volume and code of cable fault locations.

Fault Location	Sample Data Size	Identification Tags
0–20 m of the cable measuring section	123	1
20–40 m of the cable measuring section	123	2
40–60 m of the cable measuring section	123	3
60–80 m of the cable measuring section	123	4
80–100 m of the cable measuring section	123	5

**Table 5 sensors-23-00684-t005:** Comparison of results from different models.

Model	Classification and Location Training Time/s	Classification Accuracy/%	Location Accuracy/%
DBN	236.84	99.27	100
CNN	342.73	99.35	100
LSTM	220.46	97.92	99.67
ANN	195.62	94.33	96.29
SVM	5.06	89.76	91.57

## Data Availability

Not applicable.

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
