# Peer review of "Fault Identification and Localization of a Time−Frequency Domain Joint Impedance Spectrum of Cables Based on Deep Belief Networks"

_sensors, 2023, doi:10.3390/s23020684_

Round 1
Reviewer 1 Report
1. In this manuscript, the innovation and characteristic of this study should be further highlighted. 2. Literature survey seem incomplete, it is suggested to review and supplement some additional references on deep learning for the cable fault type identification and localization. In addition, each paper cited in Introduction should be reviewed. 3. It should be mentioned that all the works that are not the authors research should be cited. The derivation or Reference for Equation (8)~(12) should be given. 4. The curves in Figures 4~8 are difficult to distinguish. For better readability, it is suggested to represent these curves with different linetypes. 5. The results in section 4.2. are inconsistent with those in Figure 4 and Figure 5. Please check and correct it.Author Response
Please see the attachment! Thank you!

Reviewer 2 Report
This paper proposes fault identification and localization of Time-frequency domain joint impedance spectrum of cables based on Deep Belief Networks, which has high recognition accuracy and fast detection speed. The research has certain application value.
1、Figures 4, 5, 6 and 7 are too simple to describe the change rules of the explanatory diagrams, which need to be explained in combination with theory.
2、In the second line of section 4.3,5.2 is wrong.
3、Figure 8 is not clear enough.
4、In Table 5, DBN takes longer than ANN and SVM. What are its advantages?
5、The conclusion summary is too general, so it is better to summarize it in sections.
Reviewer 3 Report
This paper applies deep learning to cable fault diagnosis and proposes a fault identification and localization of time-frequency domain joint impedance spectrum of cables based on deep belief networks method. It is of interest. However, the authors should address the following points to further improve the quality.
1. The contributions of this paper are not clear. They should be highlighted at the end of the introduction. In addition, more information on the background and problem statement should be added.
2. Why the deep belief network is used? The motivation of the proposed hybrid fault diagnosis method should be clearly introduced.
3. Detailed comparison with prior works should be added. Please give reasoning of the results and a deeper explanation.
4. The validation is a big concern in this work. How authors can avoid bias validation? Authors can perform a significant test to show the efficiency of the method.
5. Literature review on the data-based methods of fault diagnosis is limited. More recently-published papers in this field should be discussed. The authors may be benefited by reviewing more papers such as 10.1016/j.ymssp.2022.109569 and 10.1016/j.ymssp.2022.109834.
6. The linguistic quality needs improvement. I can find some typos.
7. The quality of most figures need to be improved. They are not clear.
Reviewer 4 Report
- Page 6, line 250, an mistake on the 31 references I suppose is the 21 reference ... need to be corrected
- Line 486 the number 19 is repeated and should be removed
- Please avoid placing a figure at the bottom of the section
Round 2
Reviewer 3 Report
This paper has been improved a lot after revision. However, there are still some problems need to be fixed.
1. Please check the expression of some equations, such as Eqs. (2), (8), (9), (11), etc.
2. Do you have some practical experimental data that can be used in Section 4?
3. More powerful and recently deep learning models, such as CNN, LSTM, transformer, or other deep learning-based fault classification method, should be compared in Table 5. DBN, ANN, and SVM are classical methods.
